# Effect of Heterointerface on NO_2_ Sensing Properties of In-Situ Formed TiO_2_ QDs-Decorated NiO Nanosheets

**DOI:** 10.3390/nano9111628

**Published:** 2019-11-16

**Authors:** Congyi Wu, Jian Zhang, Xiaoxia Wang, Changsheng Xie, Songxin Shi, Dawen Zeng

**Affiliations:** 1State Key Laboratory of Material Processing and Die & Mould Technology, School of Materials Science and Engineering, Huazhong University of Science and Technology (HUST), Wuhan 430074, China; wucongyi@163.com (C.W.); jian.zhang1@anu.edu.au (J.Z.); wxxjohn@163.com (X.W.); csxie@mail.hust.edu.cn (C.X.); 2State Key Lab of Digital Manufacturing Equipment and Technology, Huazhong University of Science and Technology (HUST), Wuhan 430074, China; 3Hubei Collaborative Innovation Center for Advanced Organic Chemical Materials, Hubei University, Wuhan 430062, China

**Keywords:** TiO_2_ QDs, NiO nanosheets, NO_2_, quantum size, heterointerface

## Abstract

In this work, TiO_2_ QDs-modified NiO nanosheets were employed to improve the room temperature NO_2_ sensing properties of NiO. The gas sensing studies showed that the response of nanocomposites with the optimal ratio to 60 ppm NO_2_ was nearly 10 times larger than that of bare NiO, exhibiting a potential application in gas sensing. Considering the commonly reported immature mechanism that the effective charge transfer between two phases contributes to an enhanced sensitivity, the QDs sensitization mechanism was further detailed by designing a series of contrast experiments. First, the important role of the QDs size effect was revealed by comparing the little enhanced sensitivity of TiO_2_ particle-modified NiO with the largely enhanced sensitivity of TiO_2_ QDs-NiO. Second, and more importantly, direct evidence of the heterointerface charge transfer efficiency was detailed by the extracted interface bond (Ti-O-Ni) using XPS peak fitting. This work can thus provide guidelines to design more QDs-modified nanocomposites with higher sensitivity for practical applications.

## 1. Introduction

Recently, quantum dots (QDs) have attracted great attention in various fields due to their large specific surface area, quantum size effects, and single crystal structure with higher stability [1,2,3,4,5,6,7,8,9]. Especially, as a kind of surface modifier, semiconducting QDs have been widely employed to improve the gas sensing properties of metal-oxide semiconductor (MOS)-based materials, taking advantage of QDs’ size-induced active sites and the formation of heterojunctions [10,11,12,13,14,15]. It was generally believed that the effective charge transfer between QDs and MOS contribute to the enhanced sensitivity of QDs-decorated sensing materials. For instance, CuInS_2_ QDs-modified NiO nanosheets exhibited enhanced sensitivity to NO_2_ at room temperature compared to bare NiO nanosheets [16]. The authors attributed this enhanced sensitivity to the effective heterointerface charge transfer between NiO nanosheets and CuInS_2_ QDs. However, the effect of QDs’ size on the sensing properties of nanocomposites was not involved, because the charge transfer efficiency is largely dependent on QDs’ size due to changes of the energy band structure and specific surface area in QDs with different sizes. Moreover, the widely reported reason for the enhanced sensitivity is always based on the qualitative description of the heterointerface charge transfer, lacking the corresponding evidence. In that case, the imperfection of the QDs sensitization mechanism will certainly lead to an obstacle for the further improvement of sensing properties.

NO_2_, as a major air pollutant, has caused many problems, such as acid rain, photo-chemical smog, respiratory diseases, etc. [17,18,19]. Meanwhile, chronic obstructive pulmonary disease can be monitored by measuring the concentration of NO_x_ in exhaled gases [19,20,21,22]. Therefore, it is urgent to design NO_2_ sensors with a higher sensing performance. NiO, a p-type metal-oxide semiconductor, is considered as a promising room temperature NO_2_ sensing material because of its high adsorption energy and great changes in electronic structure under the adsorption of NO_2_ gas [23,24,25]. However, in actual NO_2_ detection, bare NiO usually fails to meet the higher requirements in limit detection concentration [26,27]. Therefore, it is vital to improve the sensing properties of bare NiO by modification. TiO_2_ is an eco-friendly n-type metal oxide semiconductor with strong chemical and thermal stability [28,29]. Meanwhile, TiO_2_ can detect NO_2_ gas at elevated temperatures [30,31]. Based on the highly surface activity and appropriate energy band structures of TiO_2_ QDs, it is reasonable to believe that TiO_2_ QDs modification can improve the room temperature NO_2_ sensing performance of bare NiO.

In this work, NiO nanosheets modified with different amounts of TiO_2_ QDs were firstly prepared. It was found that the room temperature NO_2_ sensing performances of nanohybrids were significantly enhanced compared to bare NiO nanosheets, indicating the important role of TiO_2_ QDs. Furthermore, to further clarify the quantum size effects, NiO nanosheets modified with different sizes of TiO_2_ nanoparticles were subsequently prepared. The results show that the sensitivity of the nanohybrids increased with both the increase in the quantity of TiO_2_ nanoparticles of the same particle size and the decrease of the TiO_2_ nanoparticle size. It was believed that the TiO_2_ QDs with an appropriate energy band structure could form effective heterojunctions with the NiO nanosheet, facilitating effective charge transfer. To confirm this, on the basis of different nanocomposites with different amounts of TiO_2_ QDs or TiO_2_ nanoparticles, interface bonds (Ni-O-Ti) were extracted by XPS fitting. It was found that the variations of the interface bond were in accordance with the sensitivity, directly indicating the enhanced heterointerface charge transfer efficiency in TiO_2_ QDs-modified NiO nanosheets with an optimal ratio due to the larger number of interface bonds. We hope this work could help deepen the understanding of the sensing mechanism of QDs-based nanohybrids and to design more QDs-modified nanohybrids with higher sensing performances.

## 2. Experimental

### 2.1. Sources of TiO_2_ Nanoparticles of Different Sizes

On the basis of the process reported by Zaharescu [32], we successfully synthesized TiO_2_ QDs. Firstly, n-butanol (AR; 99.5%; 150 mL) was slowly poured into a three-necked flask (250 mL). Secondly, on a magnetic stirrer, concentrated nitric acid (65.0%~68.0%, 0.55 mL) and 0.81 mL of deionized water were slowly dropped into the three-necked flask at room temperature. Thirdly, nitrogen was introduced into the three-necked flask, and titanium butoxide (AR; 98.0%; 6.8 mL) was dropwise added to the solution under stirring for 30 min. Finally, the solution was transferred to a reactor at 200 °C for 8 h, then naturally cooled to room temperature, and the nominal concentration of the prepared TiO_2_ QDs solution was 0.21 mg/mL.

TiO_2_ nanoparticles with diameters of 15 and 30 nm were purchased from Shanghai XinZuan Alloy Material Co. (AR; ≥99.9%)

### 2.2. Synthesis of the NiO Nanosheets

As shown in Appendix A, 6 g of nickel chloride (AR; ≥98.0%) were first dissolved in 300 mL of distilled water. Secondly, the solution pH was adjusted to 10.0 by adding ammonia solution (AR; 28%). Thirdly, the solution was transferred to a reactor at 160 °C for 6 h, and then naturally cooled to room temperature. Finally, the precipitate was washed three times with distilled water and ethanol by centrifugation at 8000 rpm, and the sample was obtained after drying at 80 °C.

### 2.3. Synthesis of TiO_2_ Nanoparticle-Decorated NiO Nanosheets

Firstly, 0.001 mol NiO nanosheets were dispersed in 50 mL of n-hexane. Secondly, different volumes of TiO_2_ QDs solution (1, 2, 5, and 10 mL) were added respectively in the prepared NiO nanosheets solution, and treated by ultrasonic waves for 30 min to ensure adequate mixing. Finally, the precipitate was washed three times with water and distilled by centrifugation at 3000 rpm, and the samples were obtained after drying at 80 °C. The NiO nanosheets were added to different volumes of TiO_2_ QDs solution named 1TiO_2_QDs-NiO, 2TiO_2_QDs-NiO, 5TiO_2_QDs-NiO, and 10TiO_2_QDs-NiO, respectively. For comparison, bare NiO nanosheets and bare TiO_2_ QDs were also prepared for testing.

Firstly, different quantities of TiO_2_ nanoparticles (0.001, 0.002, 0.005, 0.010, 0.050, 0.100, and 0.200 g) with diameters of 15 and 30 nm, respectively, were added to 0.01 mol bare NiO nanosheets for full grinding. Secondly, sintering processes were performed in an electric oven, which was maintained at a temperature of 200 °C for 30 min and then naturally cooled to room temperature. Finally, the NiO nanosheets modified with different quantities of TiO_2_ nanoparticles were obtained, and denoted as 1TiO_2_15-NiO, 2TiO_2_15-NiO, 5TiO_2_15-NiO, 10TiO_2_15-NiO, 20TiO_2_15-NiO, 50TiO_2_15-NiO, 100TiO_2_15-NiO, 1TiO_2_30-NiO, 2TiO_2_30-NiO, 5TiO_2_30-NiO, 10TiO_2_30-NiO, 20TiO_2_30-NiO, 50TiO_2_30-NiO, and 100TiO_2_30-NiO, respectively.

### 2.4. Characterization

Transmission electron microscopy (TEM, Tecnai G2F20 U-TWIN, 200 kV, Hillsboro, America) was carried out to observe the morphologies of the samples. The phase structure and chemical compositions were analyzed by X-ray diffraction (XRD, X’pert PRO, PANalytical B.V., Almelo, The Netherlands, CuKα radiation with λ = 0.15406 nm) from 10° to 90°. The valence band spectra and surface compositions were tested by X-ray photoelectron spectroscopy (XPS, VG Multilab 2000X, Thermo Fisher Scientific Inc., Waltham, MA, USA), and the binding energies were referenced to the C 1s peak at 284.6 eV. Tested in the ultrahigh vacuum (UHV) container, He Ι excitation (21.2 eV) was used to detect the ultraviolet photoelectron spectroscopy (UPS) spectrum.

### 2.5. Fabrication of Gas Sensors

Based on previous research in our laboratory [33,34], the suspension was prepared to a solid membrane by thick film technology and its gas sensitivity was tested. Firstly, a gold interdigital electrode was printed on an alumina ceramic sheet by the screen printing process (Appendix A). Secondly, the suspension was prepared by mixing 20 mg of prepared powder with 0.2 mL of ethanol. Thirdly, two microliters of the prepared suspension were dropped on the gold interdigital electrode. Finally, the fabricated sensors were naturally air-dried and were then placed in a thermostat at 80 °C for 24 h.

### 2.6. Gas Sensing Performance Testing

In our previous work, we described the test platform in detail [35,36], which is equipped with a data acquisition system and gas distribution device. Before the test, the gas sensors were installed in the test chamber, and then rinsed with dry air for 30 min. Afterwards, different concentrations of NO_2_ were obtained by mixing high-concentration NO_2_ with dry air, and the general flow was controlled at 1 L/min. The detailed testing process was as follows: (1) Injected in dry air for 100 s; (2) injected in testing gas for 400 s; and (3) injected in dry air for 500 s. At room temperature, complete desorption of NO_2_ molecules from the nickel matrix composites is impossible. In order to ensure the repeatability and reliability of the test, before each test, samples were heated at 80 °C for 24 h to eliminate residual NO_2_ molecules and restore them to the original state.

In this study, the gas sensing performance was evaluated by the response value (*S*), and the response value was evaluated by the following formula:(1)$$S\;=\;\frac{R_{o}{-R}_{g}}{R_{o}},$$
where *R_o_* and *R_g_* represent the resistance values of the sensors in air and the target gas, respectively.

## 3. Results and Discussion

### 3.1. Gas Sensing Performances

For the TiO_2_ QDs-modified NiO nanosheets, the dynamic sensitivity–recovery curves and responses versus NO_2_ concentration (5–60 ppm) at room temperature are shown in Figure 1a,b. The results show that, compared with the bare NiO nanosheets, the sensitivity to NO_2_ of TiO_2_QDs-NiO nanohybrids were significantly improved. More importantly, the sensitivity values increased with the increase of the quantity of TiO_2_ QDs, and decreased slightly when the QDs solution was added in excess of 5 mL. The responses of the nanohybrids modified by different sizes of TiO_2_ nanoparticles to 60 ppm NO_2_ at room temperature are shown in Figure 1c. The results show that the responses of the nanohybrids increased with the diminution of the TiO_2_ nanoparticle size. At the same time, with the increase of the TiO_2_ content of the same nanoparticle size, the response value first increased and then decreased too. Samples 5TiO_2_QDs-NiO, 20TiO_2_15-NiO, and 50TiO_2_30-NiO obtained the maximum response in their respective sample populations, and the response values are 18.5, 4.4, and 2.7, respectively.

As shown in Appendix A, to observe the selectivity of the prepared TiO_2_QDs-NiO nanohybrids, different kinds of gases, including NO_2_, n-butyl alcohol, CO, benzene, NH_3_, H_2_, and isoprene, were detected. Among these gases, sample 5TiO_2_QDs-NiO exhibited the highest response to NO_2_, which proves that the prepared nanohybrids have excellent electivity. Furthermore, the gas sensitivity of 5TiO_2_QDs-NiO at 60 PPm to NO_2_ was assessed to test the repeatability of the material’s gas sensitivity performance, and the dynamic sensitivity–recovery curves are shown in Appendix A. The results show that the prepared nanohybrids have excellent repeatability.

### 3.2. Structure Characterization

Are shown in Appendix A, the XRD patterns of the NiO nanosheets and the TiO_2_ QDs-modified NiO nanosheets are indexed to the cubic NiO (JCPDS, ICDD no. 78-0423). No characteristic peaks for impurity were observed, which indicates that the prepared samples have high purity. At the same time, the diffraction peaks of TiO_2_ are not shown in the patterns due to the low quantity of TiO_2_. Furthermore, to illustrate the QD concentrations of the TiO_2_QDs-NiO nanohybrids, the XPS spectra were also employed. As shown in Appendix A, with the increase of the quantity of TiO_2_ QDs, the concentration of Ti 2p of the samples also increased. The XRD patterns of 20TiO_2_15-NiO and 50TiO_2_30-NiO obtained the maximum response in their respective sample populations, and are shown in Appendix A. Excluding the XRD peaks of cubic NiO (JCPDF No. 08-0237) labeled with the hollow circle, all the other diffraction peaks marked by the solid circle are designated to the anatase TiO_2_ (JCPDS No. 21–1272), suggesting that the nanohybrids were successfully prepared and do not contain obvious impurities.

In order to observe the microstructure information of the prepared samples, TEM was utilized. As shown in Figure 2b,d, the TEM images show that the bare NiO is composed of nanosheets containing micropores, the synthetic TiO_2_ QDs are nanoscale black dots with a uniform size distribution, and the TiO_2_ QDs and TiO_2_ nanoparticles are uniformly separated on the surface of the NiO nanosheets. As shown in Figure 2e,f, the purchased 15 and 30 nm TiO_2_ particles were not significantly different from the actual measured size. Meanwhile, the lattice spacings were measured as 0.35 nm in the high-magnification TEM images (Figure 2d,f), attached to the (101) planes of anatase TiO_2_. In summary, the particle size of the same specification was found to be uniform, and the particle size of different specifications was quite different, which is suitable for analyzing the size effect of QDs by the difference in size.

### 3.3. Energy Band Structure of the TiO_2_ QDs

It is well known that the preparation process of QDs will affect the properties of QDs. In order to elaborate the influence of TiO_2_ QDs on the gas sensing properties of NiO nanosheets, it was necessary to determine the energy band structure of the TiO_2_ QDs. As shown in Figure 3, ultraviolet photoelectron spectroscopy (UPS), valence band XPS spectrum, and UV-vis spectroscopy were performed to calculate the energy band structure of TiO_2_ QDs. As shown in Figure 3a, the E_vac_ is 21.19 eV above the spectrum of the cutoff energy [37], and the work function of the prepared TiO_2_ QDs is 3.365 eV. Meanwhile, the valence band of the TiO_2_ QDs is shown in Figure 3b, where the position of the top of the valence band is close to the crossing point of the linear fitting for the light emission front of the valence band [38], and the maximum position of TiO_2_ QDs is 1.79 eV lower than the Fermi energy level. The ultraviolet visible (UV-vis) absorption spectrum of TiO_2_ QDs is shown in Figure 3c, and the band gap of the TiO_2_ QDs is 3.81 eV. Through the above analysis, the energy band structure of TiO_2_ QDs is shown in Figure 3d.

### 3.4. Sensing Mechanism

NiO is a p-type metal-oxide semiconductor and the hole (*h*^+^) is the main carrier, which is produced by the ionization of a neutral Ni vacancy ($V_{Ni}^{\times }$), (Equation (2)) [35,39]. During gas sensitivity measurement, NO_2_ gas molecules first adsorb on the surface of NiO, as expressed by (Equation (3)). After that, the NO_2_ gas molecules obtain electrons from a doubly charged nickel vacancy ($V_{Ni}^{\prime \prime }$) and form a singly charged nickel vacancy ($V_{Ni}^{\prime }$) (Equation (4)). Meanwhile, the consumption of $V_{Ni}^{\prime \prime }$ can promote the ionization of $V_{Ni}^{\times }$ and generate more holes that help to increase the conductivity of the nanohybrid. The reactions on the surface of the nanohybrid can be expressed as follows:(2)$$V_{Ni}^{\times }↔V_{Ni}^{\prime \prime }\;+\;2h^{+},$$
(3)$${NO}_{2}(gas)\to {NO}_{2}(ads),$$
(4)$${NO}_{2}(ads)+\;V_{Ni}^{\prime \prime }\to NO_{2}^{-}\;-\;V_{Ni}^{\prime }.$$

As for the TiO_2_ QDs-decorated NiO nanosheets, compared to the bare NiO nanosheets, the response versus NO_2_ at room temperature is significantly improved, while the response of TiO_2_ QDs versus NO_2_ is very weak (Appendix A), indicating that the improvement of the gas sensitivity performance was caused by the interaction of materials. Furthermore, in order to detail the effect of QDs on the gas sensing performances, large TiO_2_ nanoparticles-modified NiO nanosheets (15, 30 nm) were also prepared. As shown in Figure 1c, with an increased quantity of TiO_2_ of the same nanoparticle size, the response value first increased and then slightly decreased. Meanwhile, the response of the nanohybrids increased with the diminution of the TiO_2_ nanoparticle size. Based on the energy band structure of the n-type TiO_2_ QDs, it was reasonable to believe that the variation of heterointerfaces helped to enhance the sensitivity. As shown in Figure 4, the XPS spectra of the nanohybrids can be deconvoluted to six peaks, coinciding with the peaks of Ni 2p_1/2_ and Ni 2p_3/2_. Of particular note, compared with NiO nanosheets, the binding energies for the Ni 2p of the TiO_2_-modified nanohybrids shifted to higher values [40,41]. More importantly, as shown in Figure 4a,b, for the TiO_2_QDs-NiO nanohybrids, the shift value first increased and then decreased with the increase of the TiO_2_ QDs content, which is consistent with the change of the response value too. Meanwhile, as shown in Figure 4c,d, for the samples 5TiO_2_QDs-NiO, 20TiO_2_15-NiO, and 50TiO_2_30-NiO with the optimal performance in their respective sample populations, the shift value increased with the diminution of the TiO_2_ nanoparticle size, which is consistent with the change of the response value too. The shift of the binding energy is mainly due to the difference in the electronegativity of ions and the interfacial electron transfer between the NiO nanosheets and the TiO_2_ nanoparticles [42]. Compared tto the energy band structure of the NiO nanosheets and the TiO_2_ QDs [43,44,45], as shown in Figure 5a, in order to achieve the equalization of the Fermi levels, electrons are transferred from TiO_2_ QDs to NiO nanosheets.

Furthermore, the change of the heterointerface can be judged by the change of the interfacial bond content. As shown in Figure 4e, the O1s high-resolution XPS spectra of the sample can be deconvoluted into four peaks, with binding energies of 528.8, 529.8, 530.6, and 532.3 eV [46,47,48,49]. The peak at 528.8 eV can be ascribed to the lattice oxygen in NiO; the peak at 529.8 eV is identified as the Ni-O-Ti bond [33,50]; the peak at 530.6 eV can be ascribed to the defective oxygen in NiO and the crystal lattice oxygen in TiO_2_ [51,52]; and the peak at 532.2 eV can be ascribed to the hydroxyl oxygen and water molecules on the surface of the mesoporous NiO. As shown in Figure 4f and Appendix A, for the TiO_2_QDs-NiO nanohybrids, the peak area ratio of the Ni-O-Ti bond first increased and then decreased with the increase of the TiO_2_ QDs content, which is consistent with the change of the response value. Meanwhile, as shown in Appendix A, for the samples 5TiO_2_QDs-NiO, 20TiO_2_15-NiO, and 50TiO_2_30-NiO with the optimal performance in their respective sample populations, the peak area ratio of the Ni-O-Ti bond increased with the diminution of the TiO_2_ nanoparticle size, which is consistent with the change of the response value too. These results clearly show that TiO_2_ QDs could offer a larger number of heterointerfaces than large TiO_2_ nanoparticles of an equivalent mass, while the increase in the quantity of TiO_2_ nanoparticles of the same particle size would also increase the content of heterointerfaces, which helps to improve the charge transport efficiency.

As shown in Figure 5b, when the NiO nanosheets was contacted with the TiO_2_ QDs, compared to large TiO_2_ nanoparticles, a larger number of heterointerfaces were formed, which can greatly improve the transfer efficiency of electronics. Upon exposure to NO_2_, the ultra-efficient electron transfer based on more heterointerfaces from TiO_2_ QDs to NiO nanosheets could significantly promote the ionization reaction of the neutral Ni vacancy, and more quantities of charged Ni vacancies would participate in the NO_2_ sensing reaction (Equation (4)). Hence, as shown in Figure 5c, for the surface band bending, the TiO_2_ QDs-modified NiO nanosheets, large TiO_2_ nanoparticles-modified NiO nanosheets, and bare NiO nanosheets are in a descending order, which is consistent with the change in gas sensitivity.

Furthermore, the quantity of holes in the NiO nanosheets reduced with the electrons transferred from the TiO_2_ QDs to the NiO nanosheets, which resulted in a reduction of the initial conductivity of the nanohybrids (Appendix A). According to the definition of response, the reduction of the initial conductivity in the improved the response value of the nanohybrid. For the NiO nanosheets modified with different quantities of TiO_2_ nanoparticles of the same particle size, the room temperature sensitivity increased with the increase of the TiO_2_ nanoparticles. When the quantity of TiO_2_ nanoparticles further increased, the sensitivity of the nanohybrids decreased slightly. The possible reason is that the heterointerfaces increased with the increase of the TiO_2_ nanoparticles, but further increasing the quantity of TiO_2_ nanoparticles could occupy the reactive site for gas sensing, resulting in agglomeration and leading to a slight decrease of the sensitivity [16].

## 4. Conclusions

In summary, by comparing the sensitivity of NiO nanosheets modified with different contents of TiO_2_ QDs and different sizes of TiO_2_ nanoparticles, the effect of QDs on the gas sensing performances of nanohybrids was discussed in detail from the two dimensions of concentration and nanoparticle size, taking the heterointerface as the core. For the NiO nanosheets modified by different contents of TiO_2_ QDs, the binding energy shift value and the peak area ratio of the Ni-O-Ti bond first increased and then decreased with the increase of the TiO_2_ QDs content, which was consistent with the change of the respond value. Meanwhile, for the samples with the optimal performance in their respective sample populations, the binding energy shift value and the peak area ratio of the Ni-O-Ti bond increased with the diminution of the TiO_2_ nanoparticle size, which was consistent with the change of the response value too. It is believed that the increase in the quantity of TiO_2_ QDs will increase the content of heterointerfaces, while TiO_2_ QDs could offer more heterointerfaces than large TiO_2_ nanoparticles. Moreover, according to the definition of the response value, reducing the initial conductance helps to improve the sensitivity of the materials. Through this work, we can better understand the impact of quantum size on the QDs’ sensitization mechanism on the variation of heterointerfaces.

## Figures and Tables

**Figure 1 nanomaterials-09-01628-f001:**
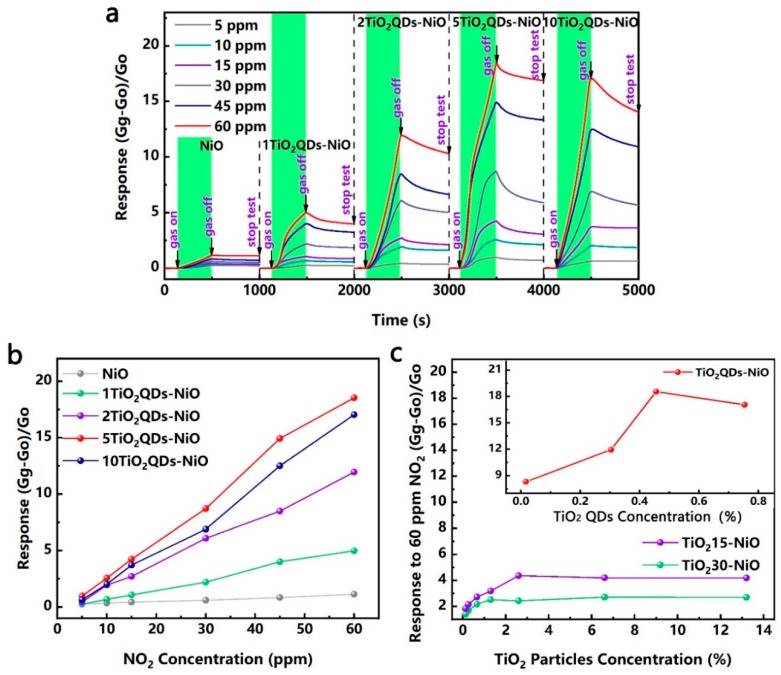
(**a**) Dynamic sensitivity–recovery curves and (**b**) responses of the bare NiO nanosheets and the NiO nanosheets modified with different quantities of TiO_2_ QDs to different concentrations of NO_2_ at room temperature; (**c**) Responses of NiO nanosheets modified with different sizes of TiO_2_ nanoparticle to 60 ppm NO_2_ at room temperature.

**Figure 2 nanomaterials-09-01628-f002:**
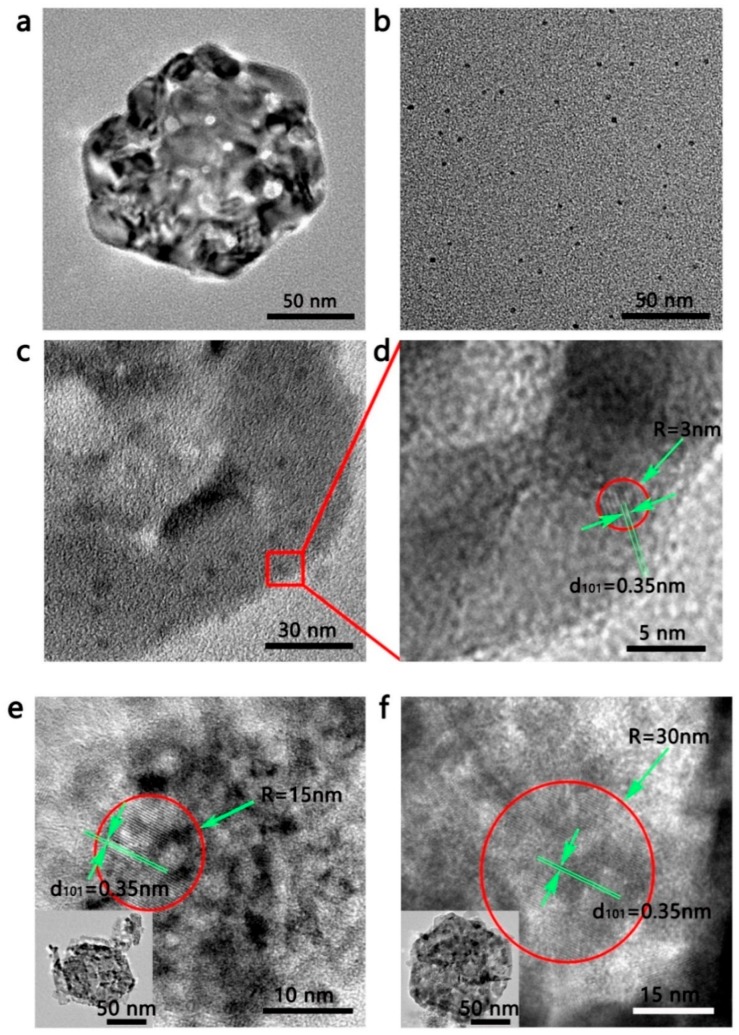
TEM images of (**a**) bare NiO nanosheets, and (**b**) bare TiO_2_ QDs. Low- and high-magnification images of (**c**,**d**) TiO_2_QDs-NiO, (**e**) TiO_2_15-NiO, and (**f**) TiO_2_30-NiO.

**Figure 3 nanomaterials-09-01628-f003:**
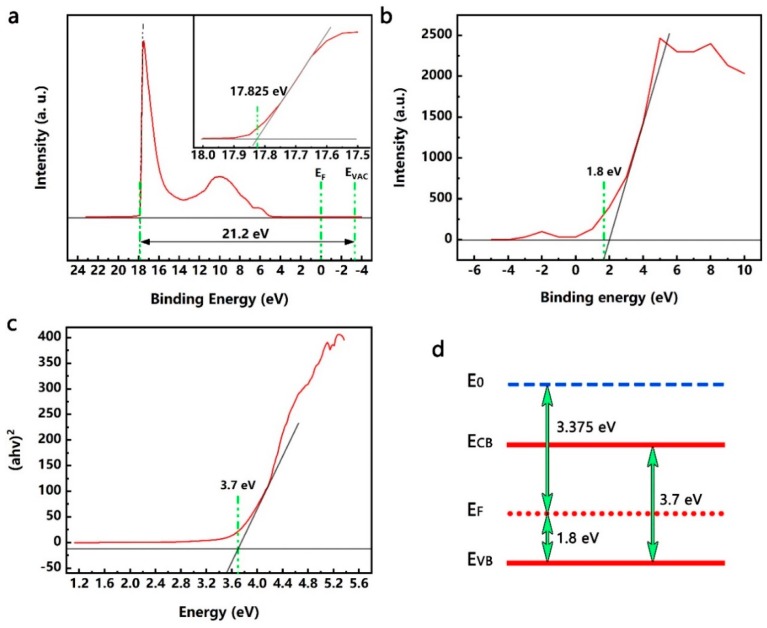
(**a**) The UPS spectra of TiO_2_ QDs; (**b**) The XPS spectrum of TiO_2_ QDs; (**c**) The plot of (αhυ)^2^ against photon energy (hυ) for TiO_2_ QDs; (**d**) The corresponding energy band structure of TiO_2_ QDs based on the UPS, XPS, and UV-vis results.

**Figure 4 nanomaterials-09-01628-f004:**
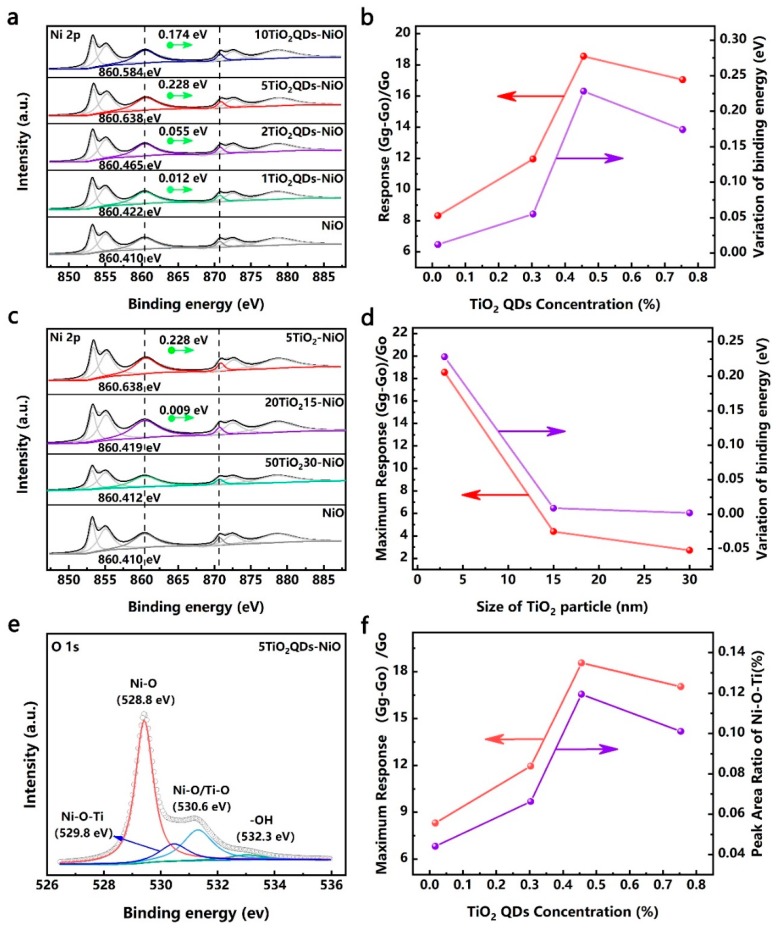
(**a**) Ni 2p spectra of the bare NiO and the TiO_2_QDs-NiO nanohybrids. (**b**) With the increase of the TiO_2_ QDs content, a comparison between the variation of the responses to 60 ppm NO_2_ and the variation of the binding energy shift values is shown. (**c**) Ni 2p spectra of the bare NiO and the 5TiO_2_QDs-NiO, 20TiO215-NiO, and 50TiO230-NiO. (**d**) With the increase of the TiO_2_ nanoparticle size, a comparison between the variation of the maximum responses to 60 ppm NO_2_ and the variation of the binding energy shift values is shown. (**e**) O 1s spectra of the 5TiO_2_QDs-NiO. (**f**) With the increase of the TiO_2_ QDs content, a comparison between the variation of the responses to 60 ppm NO_2_ and the variation of the peak area ratio of Ni-O-Ti is shown.

**Figure 5 nanomaterials-09-01628-f005:**
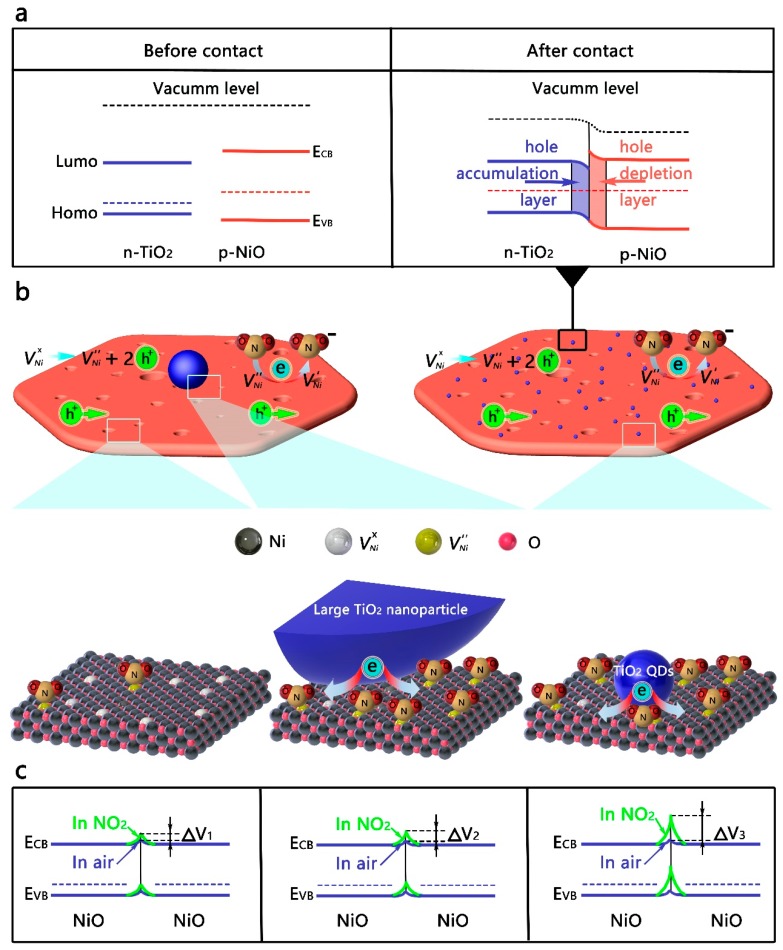
(**a**) The energy band diagram of the as-prepared samples, before and after contact. (**b**) The schematic of the interfacial interaction between the TiO_2_-NiO_2_ nanohybrids structure and NO_2_ molecules. (**c**) Comparison of the surface band bending between NiO nanosheets, large TiO_2_ nanoparticle-modified NiO nanosheets, and TiO_2_ QDs-modified NiO nanosheets.

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
