# Peer review of "Effect of Heterointerface on NO2 Sensing Properties of In-Situ Formed TiO2 QDs-Decorated NiO Nanosheets"

_nanomaterials, 2019, doi:10.3390/nano9111628_

Round 1
Reviewer 1 Report
This manuscript showcases the potential of TiO2 QD-coated NiO2 nanosheet for improved NO2 sensing. The TiO2 QD modification successfully increases the response to NO2 gas selectively, and the details of interfacial band structure is well described. However, there are several confusing information and unreferenced argument, which strongly mitigates the publication in this form. I think that further grounds for their characterization is essential for the publication in nanomaterials.
Comment 1: In line 174, the size of TiO2 QD is introduced as 3, 15, 30 nm from the TEM images. It lacks of the minimal information of size distribution such as mean and standard deviation, and it’s not very clear how the authors calculated the size of TiO2 QD from the TEM image. The 3 nm-sized TiO2 QD in Figure 2b only has TEM image showing its size. In accordance of the effect of QD size on sensor response, authors need to recalculate the TiO2 QD size.
Comment 2: In experimental section, both section 1 and section 3 are representing “Synthesis of TiO2 QDs”. In section 2.1, it seems the authors synthesize TiO2 QD by themselves, but in section 2.3, authors state that TiO2 QDs were purchased from the vendor. It should be clearly noted how they prepared each TiO2 QD.
Comment 3: In Figure 3c, the line does not seem to match (ahv)2 = 0 when they estimated the bandgap of TiO2 QD. They graph needs to be re-written and please re-calculate the bandgap.
Comment 4: In line 237, authors state that XPS peak at 529.8 eV is identified to the Ni-O-Ti bond, but the related reference 47 and 48 do not have any information about it. Please cite the proper reference to show the ground of Ni-O-Ti bond and the peaks.
Comment 5: In Figure 1c, inset graph lacks of x axis label.
Comment 6: There are lots of typos, please correct them before publication. For example, X-ray photoelectron spectroscopy in line 106.
Author Response
Comment 1: In line 174, the size of TiO2 QD is introduced as 3, 15, 30 nm from the TEM images. It lacks of the minimal information of size distribution such as mean and standard deviation, and it’s not very clear how the authors calculated the size of TiO2 QD from the TEM image. The 3 nm-sized TiO2 QD in Figure 2b only has TEM image showing its size. In accordance of the effect of QD size on sensor response, authors need to recalculate the TiO2 QD size.
Author reply:
Thanks for your meticulous comment, and your consideration is very legitimated.
The important role of QDs size effect was revealed by comparing a little enhanced sensitivity of TiO2 particle-modified NiO with largely enhanced senstivity of TiO2 QDs-NiO. For this purpose, we believe that when the size of the particles of the same specification is uniform, and the particle sizes of different specifications differ greatly, the size effect of the QDs can be well reacted.
The size of particles can be judged by the lattice fringe. [16, 50] As shown in Figure. 2b-c, the synthesized QDs are similar in size and have a diameter of about 3 nm. As shown in Figure. 2e-f, the purchased 15 nm and 30 nm TiO2 particles are not significantly different from the actual measured size.
Based on the above considerations, there is no statistical analysis of particle size in this paper. But, it is necessary to make proper explanation according to the suggestions of reviewers.
The detailed modifications are listed as below:
(Line 173) “In order to observe the microstructure information of the prepared samples, TEM was utilized. As shown in Fig. 2, the TEM images that the bare NiO are composed of nanosheets containing micropores, the synthetic TiO2 QDs are nanoscale black dots, the TiO2 QDs and TiO2 nanoparticles are uniformly separated on the surface of the NiO nanosheets. Meanwhile, the lattice spacings are measured to be 0.35 nm in the high magnification TEM images (Fig. 2d-f) is attached to the (101) planes of anatase TiO2. By measurement, the diameter of TiO2 QDs and TiO2 nanoparticles were 3nm, 15nm and 30nm, respectively.” modified to “In order to observe the microstructure information of the prepared samples, TEM was utilized. As shown in Fig. 2b-d, the TEM images that the bare NiO are composed of nanosheets containing micropores, the synthetic TiO2 QDs are nanoscale black dots with uniform size distribution, the TiO2 QDs and TiO2 nanoparticles are uniformly separated on the surface of the NiO nanosheets. As shown in Figure. 2e-f, the purchased 15 nm and 30 nm TiO2 particles are not significantly different from the actual measured size. Meanwhile, the lattice spacings are measured to be 0.35 nm in the high magnification TEM images (Fig. 2d-f) is attached to the (101) planes of anatase TiO2. In summary, the particle size of the same specification is uniform, and the particle size of different specifications is quite different, which is suitable for analyzing the size effect of QDs by the difference in size.”
Comment 2: In experimental section, both section 1 and section 3 are representing “Synthesis of TiO2 QDs”. In section 2.1, it seems the authors synthesize TiO2 QD by themselves, but in section 2.3, authors state that TiO2 QDs were purchased from the vendor. It should be clearly noted how they prepared each TiO2 QD.
Author reply:
Thanks for your suggestion and your consideration is very comprehensive.
I am sorry that I have not been able to clearly express the source of different particle sizes of TiO2 in the article. On the basis of the process reported by Zaharescu, TiO2 QDs was synthesized by ourselves (section 2.1). TiO2 nanoparticles with diameters of 15 nm and 30 nm were purchase from Shanghai XinZuan Alloy Material Co. For the misunderstanding caused by the description, we have corrected the content.
The detailed modifications are listed as below:
(Line 73) “2.1. Synthesis of TiO2 QDs.” modified to “2.1. Sources of TiO2 nanoparticles of different sizes.”
(Line 81) Add a description. “TiO2 nanoparticles with diameters of 15 nm and 30 nm were purchase from Shanghai XinZuan Alloy Material Co. (AR; ≥99.9%)”
(Line 89) “2.3. Synthesis of TiO2 QDs and TiO2 nanoparticles decorated NiO nanosheets.” modified to “2.3. Synthesis of TiO2 nanoparticles decorated NiO nanosheets.”
(Line 98) Delete “(AR; ≥99.9%, Shanghai XinZuan Alloy Material Co.)”
Comment 3: In Figure 3c, the line does not seem to match (ahv)2 = 0 when they estimated the bandgap of TiO2 QD. They graph needs to be re-written and please re-calculate the bandgap.
Author reply:
Thanks for your meticulous comment.
According to your suggestion, we fitted figure 3c again, and then confirmed other fitting data. The re-calculated bandgap is shown below.
The detailed modifications are listed as below:
(Line 198) “Fig. 3.” modified to
Fig. 3. (a) The UPS spectra of TiO2 QDs; (b) The XPS spectrum of TiO2 QDs; (c) The plot of (αhυ)2 against photon energy (hυ) for TiO2 QDs; (d) The corresponding energy band structure o TiO2 QDs based on the UPS, XPS and UV-vis results.
(Line 191) “Evac is 21.2 eV” modified to “Evac is 21.19 eV”.
(Line 192) “TiO2 QDs is 3.375 eV” modified to “TiO2 QDs is 3.365 eV”.
(Line 194) “TiO2 QDs is 1.8 eV” modified to “TiO2 QDs is 1.79 eV”.
(Line 196) “TiO2 QDs is 3.7 eV” modified to “TiO2 QDs is 3.81 eV”.
Comment 4: In line 237, authors state that XPS peak at 529.8 eV is identified to the Ni-O-Ti bond, but the related reference 47 and 48 do not have any information about it. Please cite the proper reference to show the ground of Ni-O-Ti bond and the peaks.
Author reply:
Thanks for your thoughtfulness. According to your suggestion, we have added relevant references.
It should be noted that direct evidence of the location of Ti-O-Ni bond is lacking. Since the location of lattice oxygen in NiO (528.8 eV) and lattice oxygen in TiO2 (530.6 eV) are certain, according to the rule of interface bond position, [50, 51] the Ti-O-Ni bond is necessarily located between the two. In addition, the specific position of the interface bond in the article is determined by setting the possible location of the interface bond and the constraint relationship between the samples. (Figure S5) In summary, it is credible that 529.8 eV is identified to the Ni-O-Ti bond.
The detailed modifications are listed as below:
(Line 245) Add references [50, 51]
Comment 5: In Figure 1c, inset graph lacks of x axis label.
Author reply:
Thanks for your meticulous comment.
According to your suggestion, we labeled the X-axis in the inset graph.
The detailed modification is listed as below:
(Line 149) “Fig. 3.” modified to
Comment 6: There are lots of typos, please correct them before publication. For example, X-ray photoelectron spectroscopy in line 106.
Author reply:
Thanks for your thoughtfulness. According to your suggestion, we have carefully revised the language of the manuscript.
The detailed modifications are listed as below:
(Line 108) “X-ray photoelectron spectroscopy” modified to “X-ray diffraction”.
(Line 75) “pour” modified to “poured”.
(Line 79) “transfer” modified to “transfered”.

Reviewer 2 Report
Dear Authors,
the study is well designed and report some knowledge’s regarding the effect of QD size and amount in gas sensing properties of NiO to NO2 gas.
My suggestions can be summarized as follow:
Some improvement in the English language and style is neede Comparison to the existing literature is missing. Please consider to discuss the following papers: Light-Activated Sub-ppm NO2 Detection by Hybrid ZnO/QD Nanomaterials vs. Charge Localization in Core-Shell QD, doi: 10.3389/fmats.2019.00231 Nano-Micro Lett. (2015) 7(2):97–120, DOI 10.1007/s40820-014-0023-3 Optical absorption spectra, which are simple measurement to be obtained, of different nanomaterials are missing. Could you include them into this paper. 5TiO2QDs-NiO, 20TiO215-NiO, and 50TiO230-NiO What bothers me most is the missing part of the graphs showing the sensor recovery signal. It is important to include them in the document. The performance signal of the sensor should show:(a) Signal in the absence of gas
(b) Signal in the presence of different gas concentrations
(c) Signal after the gas flow has stopped.
These graphs will give a better idea of the sensor response time as well as the sensor recovery. If drift is observed, this phenomenon must be improved by other groups. But if you exclude it, you give the impression that some information is hidden. This is not the purpose of open scientific articles.
Author Response
#Reviewer
Author reply:
Thanks for your valuable advice.
According to your suggestion, after carefully reading the relevant articles, we believe that they help to improve the readability of this article and add them to the article.
Thanks for your meticulous comments to the sensor signal. So far, the NiO-based NO2 sensors have not been fully recovered at room temperature. Therefore, we set up a test procedure to standardize the test results of different composite materials, including baseline stability conditions, time to inject target gas, time to inject dry air, recovery conditions, etc. (section 2.6) In addition, previous experiments [33, 34] were also carried out according to this test procedure, and the consistency of the experiment can better compare the performance of different sensors.
According to your suggestion, we have modified Figure 1(a) to facilitate the reader's understanding of the test process.
The detailed modifications are listed as below:
(Line 31) Add references [9]
(Line 34) Add references [15]
(Line 122) Add references [36]
(Line 149) “Fig. 3.” modified to
